# ExistenceMap-PointPillars: A Multifusion Network for Robust 3D Object Detection with Object Existence Probability Map [note 1]

**DOI:** 10.3390/s23208367

**Published:** 2023-10-10

**Authors:** Keigo Hariya, Hiroki Inoshita, Ryo Yanase, Keisuke Yoneda, Naoki Suganuma

**Affiliations:** 1Graduate School of Natural Science and Technology, Kanazawa University, Kanazawa 920-1192, Japan; 2Advanced Mobility Research Institute, Kanazawa University, Kanazawa 920-1192, Japan; ihiroki@se.kanazawa-u.ac.jp (H.I.); ryanase@staff.kanazawa-u.ac.jp (R.Y.); k.yoneda@staff.kanazawa-u.ac.jp (K.Y.); suganuma@staff.kanazawa-u.ac.jp (N.S.)

**Keywords:** 3D object detection, automated driving, sensor fusion

## Abstract

Recognition of surrounding objects is crucial for ensuring the safety of automated driving systems. In the realm of 3D object recognition through deep learning, several methods incorporate the fusion of Light Detection and Ranging (LiDAR) and camera data. The effectiveness of the LiDAR–camera fusion approach is widely acknowledged due to its ability to provide a richer source of information for object detection compared to methods that rely solely on individual sensors. Within the framework of the LiDAR–camera multistage fusion method, challenges arise in maintaining stable object recognition, especially under adverse conditions where object detection in camera images becomes challenging, such as during night-time or in rainy weather. In this research paper, we introduce "ExistenceMap-PointPillars", a novel and effective approach for 3D object detection that leverages information from multiple sensors. This approach involves a straightforward modification of the LiDAR-based 3D object detection network. The core concept of ExistenceMap-PointPillars revolves around the integration of pseudo 2D maps, which depict the estimated object existence regions derived from the fused sensor data in a probabilistic manner. These maps are then incorporated into a pseudo image generated from a 3D point cloud. Our experimental results, based on our proprietary dataset, demonstrate the substantial improvements achieved by ExistenceMap-PointPillars. Specifically, it enhances the mean Average Precision (mAP) by a noteworthy +4.19% compared to the conventional PointPillars method. Additionally, we conducted an evaluation of the network’s response using Grad-CAM in conjunction with ExistenceMap-PointPillars, which exhibited a heightened focus on the existence regions of objects within the pseudo 2D map. This focus resulted in a reduction in the number of false positives. In summary, our research presents ExistenceMap-PointPillars as a valuable advancement in the field of 3D object detection, offering improved performance and robustness, especially in challenging environmental conditions.

## 1. Introduction

Detecting surrounding objects is a fundamental function critical for enabling automated driving on public roads [1]. Particularly in urban driving scenarios, it becomes imperative to identify traffic participants in various directions and formulate motion plans [2,3] based on the prevailing traffic conditions [4,5], encompassing intersections and lane changes. Hence, there exists a pressing need to develop recognition systems that exhibit resilience in the face of fluctuations in traffic patterns and weather conditions [6]. Automated vehicles are typically equipped with a variety of sensors, including LiDAR, Millimeter-Wave Radar (MWR), cameras, and global navigation satellite systems/inertial navigation systems (GNSS/INS), as depicted in Figure 1. These sensors facilitate object recognition through a fusion of distance measurement data from LiDAR and MWR, along with image information from cameras. Numerous methods have been introduced to ascertain the positions and shapes of objects, whether in 2D or 3D space [7,8,9,10]. One notable advancement involves the real-time classification of object categories through machine learning applied to clustered 3D point cloud data obtained from LiDAR observations [7,8].

In addition to LiDAR as a standalone sensor [11], fusion methods incorporating other sensors like MWR and cameras have gained prominence in recent years, with deep learning techniques emerging as the primary approach [1,9,10].

In the realm of object recognition through deep learning, various methods employ single-type sensors, such as LiDAR [12,13,14,15,16,17,18], cameras [19,20,21,22], or a fusion of both [23,24,25,26,27]. For instance, VoxelNet [16] is a LiDAR-based 3D object detection method that employs a 3D Convolutional Neural Network (3DCNN) to transform voxel features into 2D features. However, it faces computational constraints, achieving a processing rate of only 4.4 Hz on the KITTI benchmark [28]. In contrast, another method [17] replaces the 3DCNN with sparse convolution, achieving a higher processing speed of 20 Hz on the KITTI benchmark while enhancing detection accuracy through multiscale feature extraction. Nonetheless, real-time processing remains a computational challenge. PointPillars [12] adopts a unique approach by generating a pseudo image from voxelized LiDAR point clouds in the form of vertical pillars, achieving rapid computation at 62 Hz on the KITTI benchmark [28]. This method demonstrates improved accuracy on the KITTI test dataset [28] and stands as a quintessential 3D object detection technique. Nevertheless, these methodologies encounter difficulties when detecting objects with sparse point clouds at a distance or with a limited number of point clouds [29]. As a result, fusion methods have emerged to augment point cloud data with additional features.

One such fusion method is LiDAR–camera fusion, where camera images offer rich features like RGB color and texture, distinct from the point cloud data. Various approaches have been proposed for merging camera images with LiDAR point clouds. Some methods take raw point clouds and images as inputs [23], while others incorporate point-cloud-reflected camera objects [24,25,26,27]. These fusion techniques yield improvements in detection accuracy compared to relying solely on LiDAR for object detection. However, robust detection remains challenging in scenarios where object detection in images is hindered, such as at night or during rainy conditions when visibility and image color and texture quality may be compromised. Consequently, it becomes crucial to prioritize robust object detection under all circumstances within multimodal fusion frameworks. To address these challenges comprehensively, we introduce "ExistenceMap-PointPillars" as a solution for robust object detection across diverse scenarios. Our approach focuses on utilizing object detection results obtained from a single modality, specifically 2D object detection in images. ExistenceMap-PointPillars takes a LiDAR point cloud as input and leverages object detection outcomes from other sensors and methods. Notably, it integrates a probability map that signifies the presence of objects, derived from a LiDAR-based object detection pipeline. This probability map draws upon prior information obtained through object detection from alternative sensors and methods, effectively mapping the regions where objects are likely to exist. The probability map is seamlessly integrated with the point cloud features generated during the intermediate processing stages of PointPillars [12]. The overarching goal of ExistenceMap-PointPillars is to enhance recognition performance and achieve stable detection regardless of prevailing conditions. See Table 1.

### 1.1. Related Work

In this section, LiDAR–camera fusion methods are introduced. 

One of the pioneering methods in this category is MV3D [23], which involves the integration of raw point clouds and images into a single network, thereby transforming them into a shared feature space for object detection. An alternative approach projects point cloud data onto an image plane to infuse depth information into RGB images, enhancing object classification [30]. While these methods offer the advantage of directly incorporating unprocessed data into the network for detection and classification, they are susceptible to the limitation that a malfunction in one sensor can adversely impact the overall accuracy of detection or classification, a characteristic known as early fusion. In contrast, multistage fusion methods focus on the analysis of 2D objects within images. Considerable attention has been dedicated to two-dimensional (2D) object detection and semantic segmentation in images. Conventional methods like Single Shot MultiBox Detector (SSD) [20], YOLOv3 [21], and Pyramid Scene Parsing Network [31] exhibit impressive detection accuracy, particularly under favorable visibility conditions. One notable example of a multistage fusion model is PointPainting [24], which incorporates semantic segmentation results from images. This method involves projecting the point cloud onto the image and appending class scores based on semantic segmentation outcomes. Subsequently, the feature-enriched point cloud is input into the LiDAR object detection model, resulting in enhanced detection accuracy through supplementary information within the point cloud. Another exemplar is Frustum-PointPillars [27], which applies the previously established frustum generation method to PointPillars [12]. Frustum generation leverages detection results from images to project relevant information onto the LiDAR point cloud, which is then integrated into a 3D object detection pipeline. Frustum-PointPillars [27] confines its object detection to 3D objects located within the frustum, rendering the processing more efficient. However, multistage fusion techniques involving frustum generation heavily rely on the quality of detection results in images. Under suboptimal visibility conditions, such as during night-time or rainy weather, the likelihood of encountering false negatives and false positives in images escalates [32]. In such scenarios, frustums containing genuine objects may not be generated, consequently adversely affecting 3D object detection outcomes. These multistage fusion methodologies confront the formidable challenge of ensuring robust detection regardless of the prevailing circumstances.

The proposed approach, ExistenceMap-PointPillars, offers a novel solution to this challenge by integrating previous data to construct an object existence probability map, thereby addressing the aforementioned limitations. Table 1 indicates the differences between the proposed method and other fusion methods.

### 1.2. Contributions

The key contributions of this paper are as follows:ExistenceMap-PointPillars can integrate any sensor or method with a 3D point cloud using a pseudo 2D map that shows the object existence region in the Deep Neural Network (DNN) of PointPillars;ExistenceMap-PointPillars achieves a suppression of false positives against background objects;ExistenceMap-PointPillars achieves stable recognition even when fused sensors are intercepted;ExistenceMap-PointPillars can be trained as a sensor fusion model using only LiDAR data by performing special data augmentation for pseudo 2D maps.

The remainder of this paper is organized as follows. Section 2 explains the proposed 3D object detection method with sensor fusion. Section 3 describes the implementation details, and Section 4 concludes the paper with results and future work.

## 2. Proposed 3D Object Detection

Figure 2 provides an overview of ExistenceMap-PointPillars. This novel approach is a 3D object detection network that builds upon the foundation of PointPillars [12]. It distinguishes itself by offering the capability to seamlessly incorporate any pre-detection results with a 3D point cloud through straightforward modifications. This section elaborates on the generation of a pseudo 2D map, which is constructed using image bounding boxes detected by camera images and LiDAR object tracking as preliminary detection outcomes.

The pipeline for 3D object detection, which is built upon the foundation of PointPillars, is depicted by the black dashed line at the bottom of Figure 2. It commences with the voxelization of the input LiDAR point cloud into pillars by PointPillars, subsequently resulting in the generation of a pseudo image. Following this, feature extraction is performed, and 3DBBox regression and classification are executed. In contrast, the ExistenceMap module, represented by the upper red line in Figure 2, plays a pivotal role. This module encompasses the generation of a pseudo 2D map and a feature extraction block, which are based on multiple preliminary detection outcomes from various sensors and methods. The process begins with the generation of a pseudo 2D map, a topic expounded in Section 2.1. Subsequently, in Section 2.2, we delve into the generation of pseudo 2D maps and the extraction of associated features. These features, derived from both the pseudo image and pseudo 2D map, are then concatenated along the feature channel and seamlessly merged into the 3D object detection pipeline. Section 2.3 onwards introduces the backbone and detection heads, rounding out the comprehensive overview.

### 2.1. Pseudo Image

During the processing within the pillar feature network, a pseudo image is generated from a 3D point cloud. The input 3D point cloud is first voxelized for each pillar, and a simplified version of PointNet is applied to each voxel for feature extraction. Initially, the 3D point cloud undergoes voxelization to assume a 2D grid shape denoted as (H, W). In this process, additional features are incorporated, including the distance to the arithmetic mean of all points within the pillar and the offset from the pillar’s center. Subsequently, the feature-augmented 3D point cloud is subjected to linear layer, BatchNorm, and ReLU operations to extract the 3D point cloud features (C). Ultimately, the 3D point cloud is transformed into a pseudo image, assuming dimensions of (H, W, C).

### 2.2. ExistenceMap Module

A pseudo 2D map is generated based on multiple pre-detected results. These results are represented as error ellipses, which delineate the probable existence region of objects. These error ellipses arise from the modeling of various errors, including those stemming from sensor calibration, detected object positions, and time synchronization discrepancies between sensors. The resulting pseudo 2D map comprises N channels, categorizing objects into N classes, without segregating them based on individual sensors or methods. Subsequently, the generated pseudo 2D map undergoes feature extraction and is concatenated with the pseudo image. In this paper, the pseudo 2D map is generated solely from the detection outcomes of the camera image and LiDAR object tracking information, with a specific focus on addressing errors related to sensor calibration and detected object positions. Initially, a pseudo 2D map is constructed using the results of 2D object detection within camera images. In the context of urban automated driving, the recognition of surrounding objects in all directions is paramount for making safe decisions. In this study, surrounding objects are detected as 2D image bounding boxes using multiple cameras. As illustrated in Figure 3, each image is integrated into a unified image for fusion with LiDAR data. The process involves a coordinate transformation from the image coordinate system to the LiDAR coordinate system, facilitated by the intrinsic and extrinsic parameters of each camera. Subsequently, a polar coordinate image is generated. Each pixel within the polar image corresponds to an angle around the z-axis in the LiDAR coordinate system and an angle with respect to the xy-plane. When merging into the polar image, 2D objects spanning multiple images are consolidated through the application of non-maximum suppression.

Subsequently, the position of the detected object in the bird’s-eye view, derived from the 2D bounding boxes, is estimated as outlined below and is depicted in Figure 4.

The calculation proceeds by determining the distance, denoted as $d_{\alpha }$, from the vehicle to the object. This distance is computed based on the pitch angle $\phi_{bottom}$, which corresponds to the bottom position of the detected object in the image. This calculation assumes a flat road surface and accounts for the height, denoted as *h*, from the road surface to the LiDAR. Given that the polar image lacks depth information and road slope data, we determine the intersection point between the object’s direction and an assumed flat road surface with a specified gradient. Recognizing that terrain surfaces are often inclined, we naturally consider this inclination when deriving the distance, assuming the ground is inclined at an angle of α radians. The yaw angle $\psi_{c}$ of the 2D object and the derived distance d are used to estimate the position $p_{cam}={\left(x_{o},y_{o}\right)}^{T}$ in the LiDAR coordinate system expressed by the following Equation (1):(1)$$p_{cam}={\left(x_{o},y_{o}\right)}^{T}={\left(\frac{h\cos\psi_{c}}{\tan\emptyset_{bottom}-\tan\alpha },\frac{h\sin\psi_{c}}{\tan\emptyset_{bottom}-\tan\alpha }\right)}^{T}$$

Next, an error ellipse centered at $p_{cam}$ is mapped onto a pseudo 2D map. The covariance matrix Q of $p_{cam}$ is derived from error propagation theory as follows: $p_{cam}$ is represented by the function P in Equation (2)
(2)$$p_{cam}=P\left(h,\psi_{c},\emptyset_{bottom},\alpha \right)$$

From error propagation theory, the error ∆p in $p_{cam}$ is approximated as
(3)$$∆p=\sum_{e=h,\psi_{c},\emptyset_{bottom},\alpha }\frac{\partial P}{\partial a}∆e$$

The covariance matrix Q of p, which is the expected value of the squared error, is expressed using the variance $\sigma^{2}$ as follows:(4)$$\begin{array}{cc} Q & =E\left[∆z{\left(∆z\right)}^{T}\right]\\ & =\sum_{e=h,\psi_{c},\emptyset_{bottom},\alpha }\frac{\partial P}{\partial e}E\left[∆e{\left(∆e\right)}^{T}\right]{\left(\frac{\partial P}{\partial e}\right)}^{T}\\ & =\sum_{e=h,\psi_{c},\emptyset_{bottom},\alpha }\frac{\partial P}{\partial e}\sigma_{e}^{2}{\left(\frac{\partial P}{\partial e}\right)}^{T} \end{array}$$

Finally, the covariance matrix Q of $p_{cam}$ is derived using the variances $\sigma_{x}^{2}$ and $\sigma_{y}^{2}$ in the x and y directions in the LiDAR coordinates and is expressed as follows:(5)$$Q=\left(\begin{array}{cc} \sigma_{x}^{2} & \sigma_{xy}\\ \sigma_{xy} & \sigma_{y}^{2} \end{array}\right)$$

The error ellipse is represented by $G\left(x,y\right)$ using a covariance matrix. Here, ρ is the correlation between x and y. The error ellipse size c is set to $c^{2}=$ 5.99146 with an upper probability of 0.050 from the $\chi^{2}$ test to obtain the object existence range with a probability of 95%.
(6)$$G\left(x,y\right)=\frac{1}{\left(1-\rho^{2}\right)}\left(\frac{{\left(x-x_{o}\right)}^{2}}{\sigma_{x}^{2}}-\frac{2\rho \left(x-x_{o}\right)\left(y-y_{o}\right)}{\sigma_{x}\sigma_{y}}+\frac{{\left(y-y_{o}\right)}^{2}}{\sigma_{y}^{2}}\right)$$

The error ellipse follows a normal distribution, represented by $f\left(x,y\right)$.
(7)$$f\left(x,y\right)=\frac{1}{2\pi \sigma_{x}\sigma_{y}\sqrt{1-\rho^{2}}}\exp\left\{-\frac{1}{2}G\left(x,y\right)\right\}$$

The error ellipse, as defined in the previous 2D object detection results, is then transferred onto each class channel of the pseudo 2D map to denote the object’s existence range. Following this, a pseudo 2D map is generated based on LiDAR object tracking information. This process commences with the identification of vertical wall components, determined through the positional relationship between point clouds, with an obstacle flag assigned to the corresponding point cloud. The point cloud is then classified into objects $z_{t}$. Next, the current object status ${\hat{p}}_{tracing,t/t-1}$ is predicted from the object position and velocity in the past frame. Finally, the position of an object in the current frame, $p_{tracking,t/t}$, can be estimated using the classified and predicted objects from the past using the Kalman filter expressed in Equation (8), where $K_{t}$ is the Kalman gain, which increases with low confidence of the predicted position.
(8)$$p_{tracking,t/t}={\hat{p}}_{tracing,t/t-1}+K_{t}z_{t}$$

The error ellipse from the LiDAR object tracking information is mapped onto a pseudo 2D map using the derived object positions $p_{tracking,t/t}$ and the variance with Equations (5)–(7).

It is important to note that both camera and tracking information are not mapped separately but rather incorporated into the same channel corresponding to their respective classes. By employing a consistent approach to map error ellipses for each sensor and method class, we created a pseudo 2D map of dimensions (H, W, N), aggregating multiple pieces of information. Figure 5 provides a visual representation of this pseudo-two-dimensional (pseudo 2D) map. Camera information is characterized by larger error ellipses, whereas tracking information offers more precise location data, resulting in smaller error ellipses. 

Within the ExistenceMap feature network, we conducted feature extraction on the generated (H, W, N) pseudo 2D map. This process involves a series of steps, including 3 × 3 kernel 2D convolution, BatchNorm, and ReLU, across six layers, akin to the methodology in MapFusion [33]. The number of filters was set at 16, 16, 32, 32, 64, and 64, respectively, while maintaining the assumed unchanged dimensions of H × W for the pseudo 2D map. Ultimately, this process yields a pseudo 2D map feature of size (H, W, C). In the feature aggregation phase, the feature-extracted pseudo 2D map and pseudo image were concatenated along the feature channel, generating features in the shapes of (H, W, 2C).

### 2.3. Backbone and Detection Head

The backbone and detection head are the same as those of PointPillars [12]. In the backbone, the concatenated features $\left(H,W,2C\right)$ of the pseudo image and feature-extracted pseudo 2D map are extracted. A 2D convolutional layer was applied to the features to extract multiscale features into $\left(H/2,W/2,2C\right)$, $\left(H/4,W/4,4C\right)$, and $\left(H/8,W/8,8C\right)$. All features were upsampled into $\left(H/2,W/2,4C\right)$ by applying them to the transposed 2D convolutional layer, and all features were concatenated. Finally, all feature shapes were $\left(H/2,W/2,12C\right)$. In the detection head, detection was performed using a single-shot detector (SSD) [20]. The height and height–direction positions were determined regressively using the 2D detection results.

## 3. Evaluations

### 3.1. Implementation Details

#### 3.1.1. Dataset

In this study, we conducted evaluations using datasets obtained from our experimental vehicles, which operated on Japanese public roads in the Ishikawa, Toyama, and Tokyo prefectures. These datasets comprise LiDAR point clouds and omnidirectional camera images, meticulously annotated by a professional company. The annotations in these datasets pertain to tracking objects. For training purposes, we utilized a dataset consisting of 11,674 frames, and for testing, a separate dataset comprising 2679 frames was employed. Table 2 provides a breakdown of the item counts used in our evaluation. It is noteworthy that "trailer" refers to the truck bed section of the truck, “cyclist” encompasses both motorcycles and bicycles, and "person riding it" refers to individuals operating these vehicles. Figure 6 illustrates the occlusion rate of each object category and its distance from the ego vehicle. The figure reveals that the occluded object rate exceeded 20% for each class, and the distance distribution spanned a considerable range, ranging from 10 to 90 m. Our dataset encompasses challenging scenarios, including crowded scenes, objects concealed behind trucks, and high object densities. Figure 7 provides a visual example of such a challenging scene in Tokyo for object detection, featuring numerous surrounding cars and trucks, occluded objects, and a significant presence of pedestrians. Consequently, our datasets offer substantial value for evaluating 3D object detection models in the context of automated driving scenarios.

#### 3.1.2. Condition

The model prepared for the evaluation is as follows:PointPillars with only LiDAR point clouds [12];Frustum-PointPillars [27] with weighting and filtering, only weighting, or only filtering;ExistenceMap-PointPillars with camera and tracking information, with only camera or tracking information

The settings are listed in Table 3. The location, class, and direction weights of the loss function were set to $\beta_{loc}=$ 1.0, $\beta_{cls}=$ 1.0, $\beta_{dir}=$ 0.2. The sizes of the anchor boxes in each class are listed in Table 4. The pseudo 2D map was composed of three channels divided into cars (buses, trucks, and trailers), pedestrians, and cyclists. Pedestrians and cyclists were separated because of the difficulty in their detection from a point cloud. During the testing process, image objects were detected by applying the 2D object detection model YOLOv3 [21].

#### 3.1.3. Data Augmentation

Data augmentation plays a pivotal role in enhancing the diversity of a model, and it becomes particularly crucial when generating a pseudo 2D map to enrich the information available. During the training phase, we aimed to simulate prior information based on annotated LiDAR 3D object data, and information from other sensors could be synthesized by taking into account their uncertainties. For instance, when producing camera observations, we converted LiDAR objects into polar image coordinates and computed the associated error ellipses, as previously mentioned. In this process, it is imperative to consider scenarios involving false-positive and false-negative detections. The following is how we addressed these scenarios. Reproducing False-Negative Scenes: To recreate scenes with false negatives, we randomly selected certain objects within a frame based on their occlusion and error ellipses. This approach helps generate a realistic pseudo 2D map, as depicted in Figure 8. Imitating False Positives: false positives can be emulated by introducing dummy objects, with the number and types of dummy objects contingent upon the objects present in the frame, as illustrated in Figure 8. Accounting for Extreme False Negatives: in some cases, to accommodate extreme false negatives, such as instances of machine malfunction, a pseudo 2D map devoid of error ellipses may be employed. This data augmentation strategy serves to mitigate the potential adverse impact of prior information interception or the failure to detect objects in the prior information. Importantly, this approach obviates the need to prepare additional sensor information during the training phase. The model can be developed using solely the LiDAR object dataset. Subsequently, during the testing phase, we leveraged other sensor information to generate a pseudo 2D map.

### 3.2. Results

The subsequent subsections illustrate that ExistenceMap-PointPillars effectively achieves robust object detection by integrating multiple detection results into a 3D object detection pipeline. As baseline models, we compare ExistenceMap-PointPillars with other object detection models, namely PointPillars and Frustum-PointPillars. Frustum-PointPillars encompasses preprocessing, filtering, and weighting of the LiDAR point cloud data. Filtering entails the removal of point clouds lying outside the 2D bounding boxes, while weighting involves augmenting feature values based on a Gaussian distribution corresponding to the distance from the center pixel of the bounding box for point clouds within the 2D bounding boxes. Our evaluation is based on the following criteria:Usefulness of ExistenceMap-PointPillars (EM-PP) compared with PointPillars (PP);Robustness of ExistenceMap-PointPillars (EM-PP) in comparison to Frustum-PointPillars (F-PP);The robustness of ExistenceMap-PointPillars (EM-PP) itself compared to a scenario where prior data in testing, such as detection results from sensors or other methods, are intercepted;Effectiveness of multifusion with ExistenceMap-PointPillars (EM-PP);The computational cost;Ablation study of data augmentation for pseudo 2D maps;The visualization of PointPillars (PP) and ExistenceMap-PointPillars (EM-PP) focused on detecting objects.

Table 5 presents the Average Precision (AP) values for PointPillars (PP); Frustum-PointPillars (F-PP) and cases involving only filtering (F-PP-only Filter) and only weighting (F-PP only Weight); and ExistenceMap-PointPillars trained exclusively with camera data (EM-PP trained only Camera), exclusively with tracking data (EM-PP trained only Tracking), and with both camera and tracking information (EM-PP trained Camera & Tracking) on our dataset. Table 6 provides the Average Precision (AP) values for PointPillars (PP), ExistenceMap-PointPillars trained with both camera and tracking data (EM-PP), and ExistenceMap-PointPillars that undergoes testing without any prior camera or tracking data (EM-PP (no prior data in testing)). Table 7 furnishes precision and recall values for PointPillars (PP), ExistenceMap-PointPillars trained with both camera and tracking information (EM-PP), and ExistenceMap-PointPillars undergoing testing without prior camera or tracking data (EM-PP (no prior data in testing)). Table 8 showcases precision and recall values within and outside of the error ellipse range as mapped in the pseudo 2D map, used for evaluating PointPillars (PP) and ExistenceMap-PointPillars trained with camera and tracking information (EM-PP). A score threshold of 0.25, representing the highest F-score for pedestrians and cyclists, is applied.

#### 3.2.1. Usefulness of ExistenceMap-PointPillars (EM-PP) in Comparison with PointPillars (PP)

According to Table 5, the EM-PP trained with both camera and tracking data demonstrated a notable improvement in mean Average Precision (mAP) of +4.19% when compared to PP. Moreover, this improvement was consistent across all classes. Notably, the Average Precision (AP) for pedestrians and cyclists, which are typically challenging for 3D object detection, exhibited significant gains, with increases of +2.72% and +14.81%, respectively. Table 7 highlights that EM-PP significantly enhanced precision across all classes in comparison to PP. Furthermore, Table 8 indicates that precision outside the error ellipse and recall within the error ellipse both increased for all classes. This outcome suggests a substantial reduction in false positives for background objects and an increase in true positives within the error ellipse. However, recall outside the error ellipse decreased, signifying a reduced capacity for detecting objects beyond the error ellipse.

Figure 9 visually presents the recall values for PP and EM-PP across various distances from the ego vehicle for pedestrians and cyclists. The data illustrate an overall increase in recall across all distance ranges, particularly in the near-distance range (0–30 m). This trend suggests that the error ellipse’s proximity to the ego vehicle is particularly effective in improving the true-positive rate for pedestrians and cyclists. This is primarily due to the error ellipse being smaller at closer distances from the ego vehicle, potentially leading to higher maximum existence probability values. It is important to note that the size of the error ellipse is defined by the error variance. Therefore, a higher variance setting is anticipated to result in a loss of true positives. Additionally, if the center of the error ellipse is distant from the object’s actual location, a similar effect may occur.

#### 3.2.2. Robustness of ExistenceMap-PointPillars (EM-PP) Compared with Frustum-PointPillars (F-PP)

Table 5 reveals that F-PP, F-PP only Filter, and F-PP only Weight exhibit lower mean Average Precision (mAP) values, showing a decrease of -9.80%, -9.48%, and -3.99%, respectively, when compared to PP. The extensive coverage of our dataset across a wide area and in all directions raises the risk of directly incorporating false negatives from the camera into the 3D point cloud. Furthermore, the absence of time synchronization between the camera and LiDAR in our dataset introduces a time lag, leading to a shift in the positions of objects between the sensors and an increased risk of omitting the point cloud data for these objects. A similar risk arises from potential camera or LiDAR calibration errors. Notably, F-PP records the lowest mAP, while F-PP only Weight yields the highest mAP when compared with the other two. However, it is important to note that the mAP of F-PP only Weight remains below that of PP. Consequently, this indicates that camera images directly influence 3D detection by supplying detection results to the point cloud. Conversely, Table 6 demonstrates that EM-PP, even in the absence of prior data during testing (EM-PP no prior data in testing), maintains its mAP, suggesting that EM-PP effectively mitigates the adverse impact of information from other sensors. In other words, EM-PP exhibits greater robustness compared to F-PP.

#### 3.2.3. The Robustness of ExistenceMap-PointPillars (EM-PP) Itself

When considering fusion with other sensors, it is crucial to assume that information from other sensors may be unavailable. Table 6 demonstrates that the mAP of EM-PP (no prior data in testing) remains unchanged compared to that of PP. Furthermore, Table 7 reveals that EM-PP (no prior data in testing) exhibits higher precision than PP across all classes. Additionally, the reduction in recall is limited to within -3.13%. In essence, EM-PP (with no prior testing data) effectively mitigates false positives while preserving a significant portion of true positives. Based on these results, ExistenceMap-PointPillars achieves robust detection.

#### 3.2.4. Effectiveness of Multifusion with ExistenceMap-PointPillars (EM-PP)

In Table 5, it is evident that EM-PP trained with both camera and tracking data outperforms EM-PP trained with either camera or tracking data alone, with the highest mAP values observed, especially for pedestrians and cyclists. This highlights the effectiveness of tracking information in enhancing detection accuracy, emphasizing that the fusion of camera and tracking data leads to superior detection performance. This underscores EM-PP’s capability to facilitate sensor and method fusion.

#### 3.2.5. The Computational Cost

Table 9 provides insight into the processing times of PP, F-PP, and EM-PP. F-PP delivers significantly faster processing speeds but at the expense of detection accuracy. Conversely, EM-PP achieves the aforementioned performance improvements without an increase in computational cost.

#### 3.2.6. Ablation Study of Data Augmentation for Pseudo 2D Maps

EM-PP incorporates data augmentation for pseudo 2D maps during the training process, as outlined in Section 3.1.3. This augmentation consists of three processes aimed at simulating false negatives, false positives, and extreme false negatives in prior data. Table 10 presents the ablation results of these augmentations, with and without prior data in testing. Table 10 reveals that EM-PP without FN augmentation exhibited the highest precision but the lowest recall across all classes when prior data were available. This suggests that augmenting for false negatives in prior data is effective in reducing reliance on prior data. Conversely, EM-PP without FP augmentation displayed low precision for cars and cyclists when prior data were provided, indicating that augmenting for false positives in prior data is effective in mitigating the negative impact of false positives in prior data. Across all objects, data augmentations performed with EM-PP yielding the highest F-score when prior data were available, underscoring the effectiveness of augmenting the pseudo 2D map. Table 10 also illustrates that EM-PP without FN augmentation exhibited a significant decrease in both precision and recall in the absence of prior data. Furthermore, EM-PP without extreme FN augmentation yielded a lower F-score compared to all data augmentations performed with EM-PP in the absence of prior data. These findings emphasize the importance of data augmentations for the pseudo 2D map as a crucial process in the model’s performance.

#### 3.2.7. The Visualization of PointPillars (PP) and ExistenceMap-PointPillars (EM-PP) Focused on Detecting Objects

Grad-CAM [34] is a widely recognized method for visualizing the areas where a network’s attention is concentrated. We adapted Grad-CAM for both PP and EM-PP to visualize the evidence guiding their decision-making processes. Figure 10, Figure 11 and Figure 12 showcase the detection outcomes and network-focused visualizations using Grad-CAM for PP and EM-PP trained with camera and tracking information. A score threshold of 0.25 was applied. In Section 3.2.1, the results demonstrated that EM-PP reduced the number of false positives occurring outside the error ellipse. Figure 10 illustrates a scene where false positives outside the error ellipse have been diminished. PP directs its attention not only to regions containing actual objects but also to the background, resulting in false positives. Conversely, EM-PP exhibits reduced attention outside the error ellipse, effectively suppressing false positives. These results reveal that while PP suffers from false positives due to distractions, EM-PP manages to mitigate them by focusing on the inside of the error ellipse.

In Section 3.2.1, the results indicate that EM-PP reduces the number of true positives occurring outside of the error ellipse. Figure 11 illustrates a false-negative scenario outside the expanded parking area. EM-PP exhibits an increase in false negatives in this case because there is no error ellipse around this region, causing a suppression in the network’s response, as previously shown in Figure 10. However, it is important to note that the EM-PP’s response does not entirely disappear. In other words, there remains a possibility of detecting objects in this region. To address this issue, it may be beneficial to incorporate high-definition maps and activate responses inside the parking region.

In Section 3.2.1, the results reveal that EM-PP increases the number of true positives occurring within the error ellipse. Figure 12 provides a depiction of true positives for a cyclist in a scenario where the error ellipse has been expanded. Given the limited number of points associated with the cyclist, PP struggles to detect pedestrians. Nonetheless, PP concentrates its attention on regions where cyclists are present, despite their insufficient presence for reliable detection. In contrast, EM-PP effectively identifies the cyclist and triggers a response within the error ellipse. These results demonstrate that while PP fails to detect objects using point cloud data alone, EM-PP successfully activates responses within the error ellipse and accomplishes object detection.

## 4. Conclusions

In this study, we have tackled the challenge of achieving robust recognition through multisensor fusion.

ExistenceMap-PointPillars improves mAP by +4.19% over PointPillars;ExistenceMap-PointPillars improves precision by effectively suppressing false positives for background objects;ExistenceMap-PointPillars increases recall inside the error ellipse by more than +0.63% for all classes.ExistenceMap-PointPillars enhances mAP when by fusing multiple sensors or methods;Employing comprehensive data augmentation for the 2D maps yields the highest F-score among the ablated data augmentation models.

Since the pseudo-sensor information is generated by modeling sensor uncertainties during the training process, our model can be trained as a sensor fusion model solely using LiDAR 3D object data, without the need for additional sensor information. This flexibility allows our model to be applied with various sensor types, taking into account potential sensor errors, including time synchronization discrepancies between sensors. However, it is important to note that the detection of objects outside the existence region in a pseudo 2D map is also restrained. Integrating positional contexts may prove advantageous in activating network reactions. Additionally, the increase in recall for distant objects is relatively modest. This is attributed to the likelihood that at greater distances the error ellipse tends to be larger compared to the error ellipse at closer ranges, leading to a lower maximum existence probability value. This reduction in the maximum existence probability value hampers the increase in true positives. To address this, it is essential to fuse more modalities, such as Millimeter-Wave Radar, and introduce modifications to the pseudo 2D map generation process in order to elevate the maximum existence probability value.

## Figures and Tables

**Figure 1 sensors-23-08367-f001:**
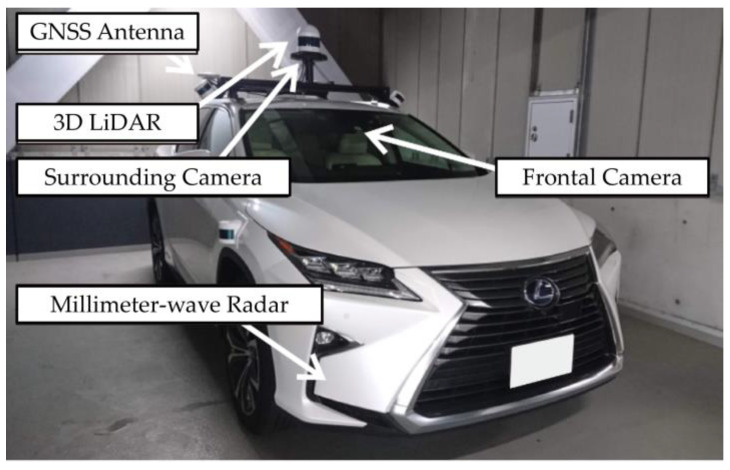
Automated vehicle.

**Figure 2 sensors-23-08367-f002:**
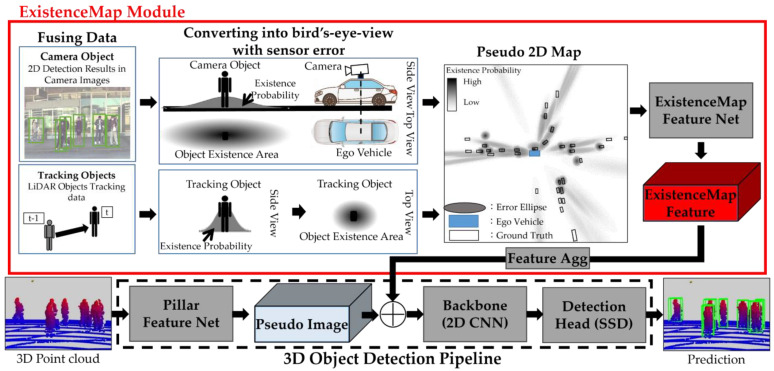
Architecture of ExistenceMap-PointPillars. The pipeline for 3D object detection based on PointPillars [12] corresponds to the black dashed line and the ExistenceMap module corresponds to the upper red line. A pseudo image generated from point cloud and pseudo 2D map features are concatenated along the feature channel and merged into a 3D object detection pipeline.

**Figure 3 sensors-23-08367-f003:**
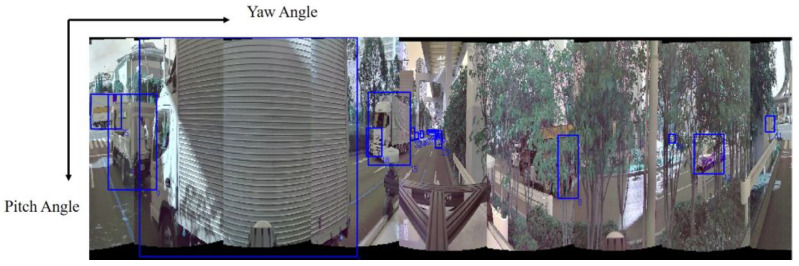
Polar image. The blue boxes are 2D detection results in a polar image. The horizontal direction represents the yaw angle and the vertical direction represents the pitch angle.

**Figure 4 sensors-23-08367-f004:**
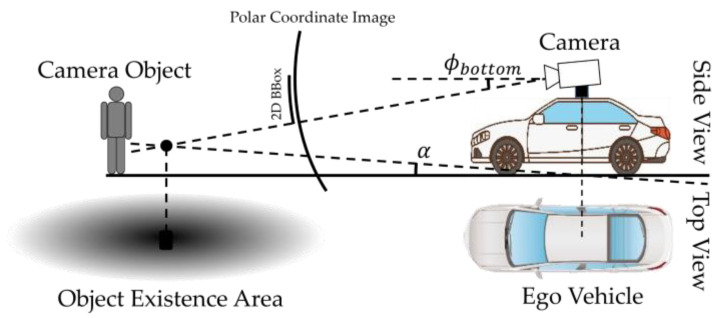
Estimation of the distance from ego vehicle to the detected object in an image.

**Figure 5 sensors-23-08367-f005:**
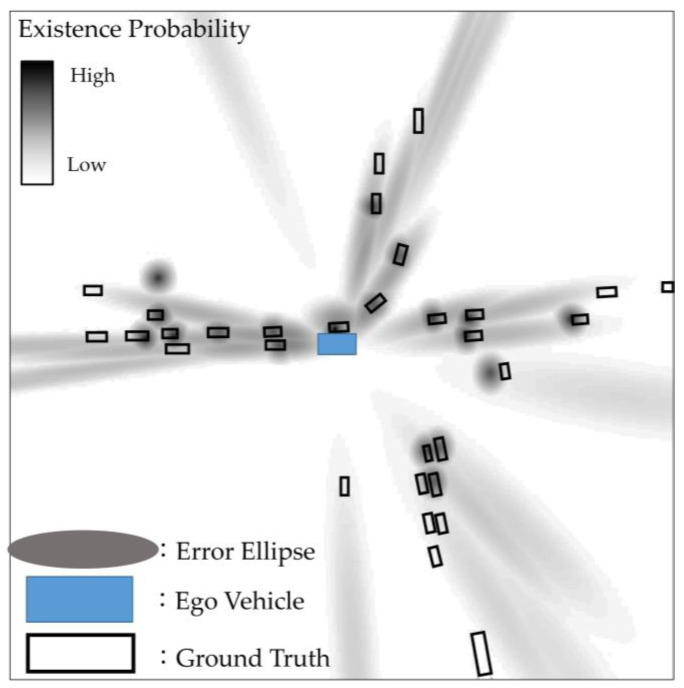
Error ellipse mapped in a pseudo 2D map using camera and tracking information.

**Figure 6 sensors-23-08367-f006:**
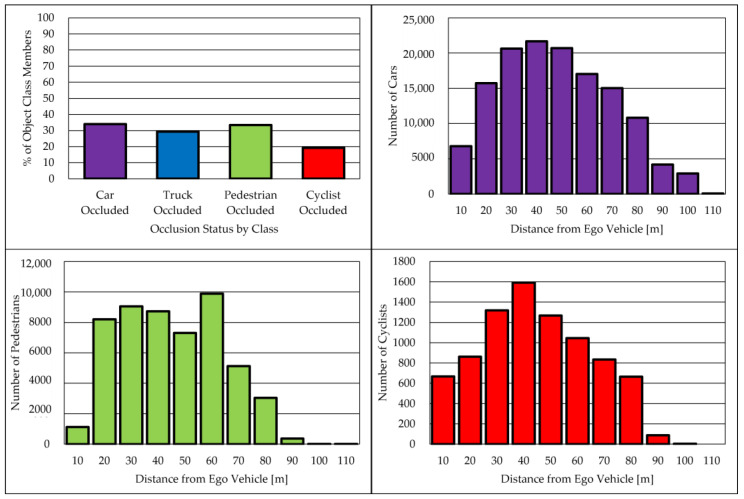
Object occlusion and distance from the ego vehicle of our dataset. The top left figure shows the occluded objects rate in each class; top right, bottom left, and bottom right ones show the distance from the ego vehicle in each class.

**Figure 7 sensors-23-08367-f007:**
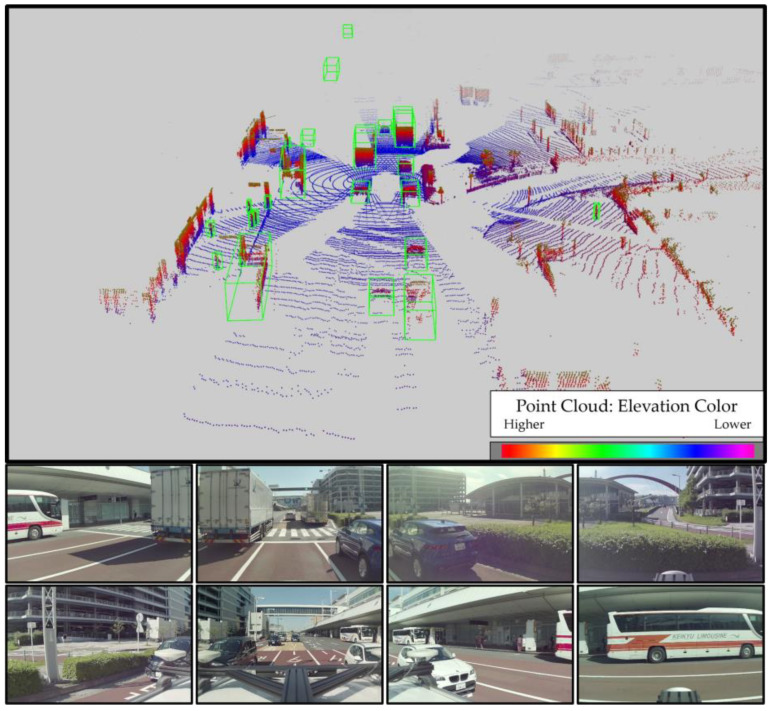
Three-dimensional point cloud and RGB images from omnidirectional cameras of our dataset. The scene was collected in Tokyo, Japan.

**Figure 8 sensors-23-08367-f008:**
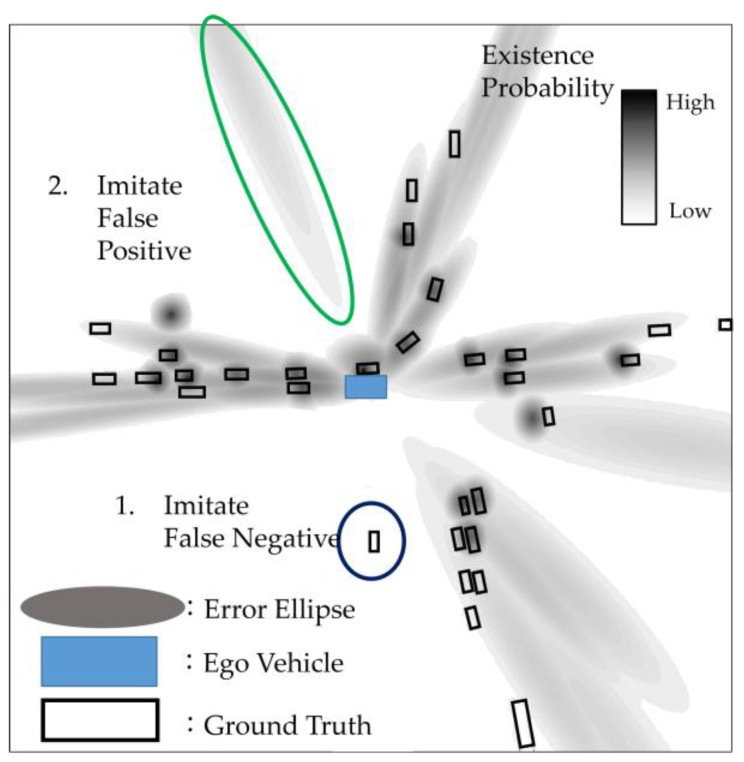
A data-augmented pseudo 2D map. The blue circle shows an imitation of a false negative in prior information, and the green circle shows an imitation of a false positive in prior information.

**Figure 9 sensors-23-08367-f009:**
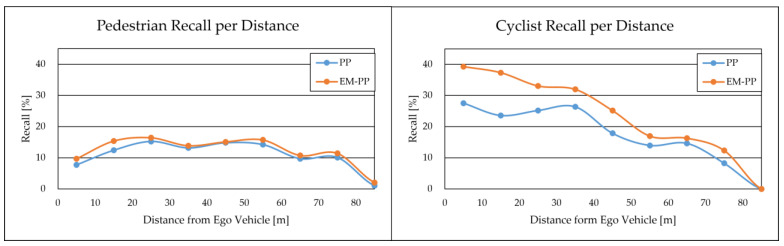
Evaluation results on our test dataset by recall (%) per distance from ego vehicle for pedestrians (**left**) and cyclists (**right**).

**Figure 10 sensors-23-08367-f010:**
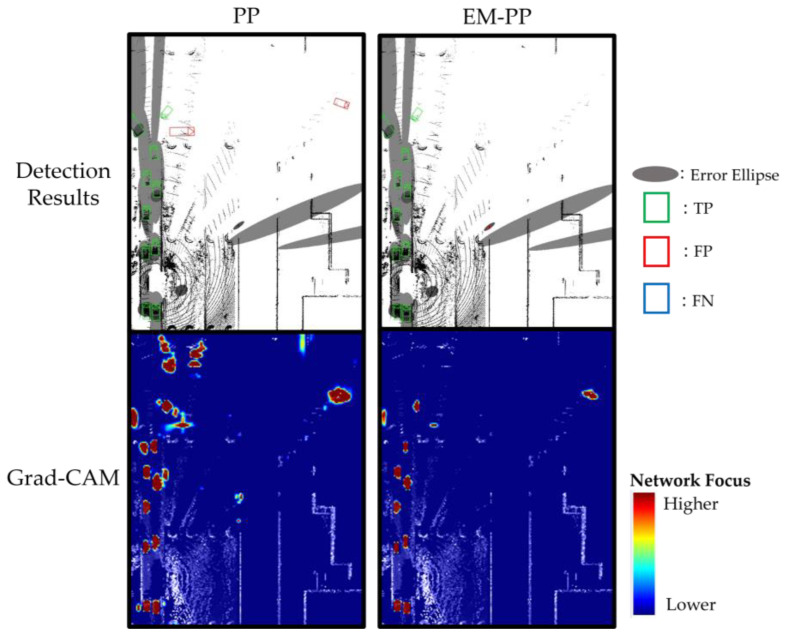
Detection results (**upper**) and where the network focused (**bottom**), visualized using Grad-CAM, in EM-PP shows decreased false positive outside of error ellipse scene. EM-PP enables the suppression of false positive by paying attention to the inside of the error ellipse.

**Figure 11 sensors-23-08367-f011:**
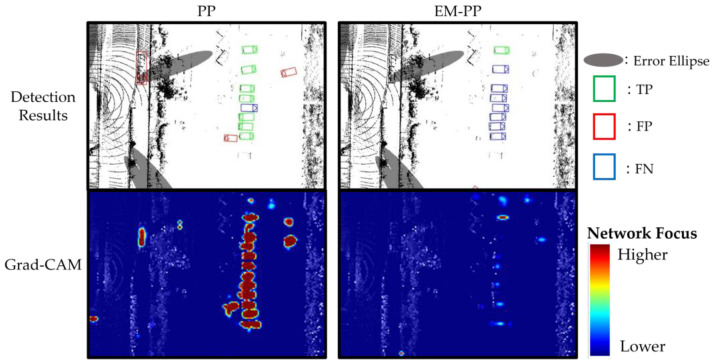
Detection results (**upper**) and where the network focused (**bottom**), visualized using Grad-CAM, in EM-PP shows increased false negative outside of error ellipse scene. EM-PP suppressed the reaction outside of error ellipses.

**Figure 12 sensors-23-08367-f012:**
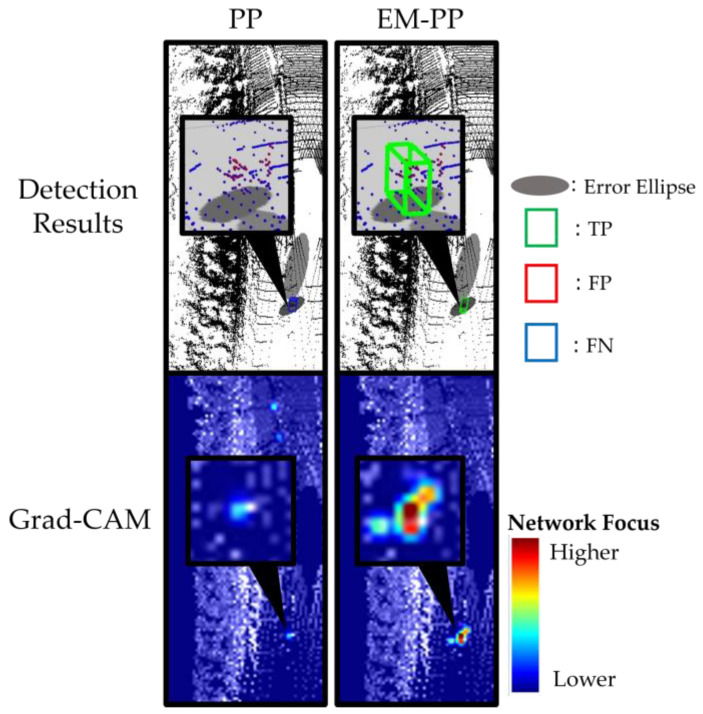
Detection results (**upper**) and where the network focused (**bottom**), visualized using Grad-CAM, in EM-PP shows increased true positives outside of error ellipse scene. EM-PP enabled the reaction inside of error ellipse to be activated and detect the cyclist.

**Table 1 sensors-23-08367-t001:** Input and 3D detection dependency on camera in MV3D [23] (early fusion), Frustum PointNet [25] and Frustum ConvNet [26], Frustum-PointPillars [27] (multistage), and our proposed ExistenceMap-PointPillars. Early fusion takes raw point cloud and camera images to a single network. Multistage fusion takes images and point cloud added features based on 2D detection results. ExistenceMap-PointPillars takes raw point cloud images and integrates an object existence map generated from 2D detection results. ExistenceMap-PointPillars could reduce the negative impact of fused sensor information.

Fusion Method	Input		3D Detection Dependency on Camera
	LiDAR	Camera	
Early (MV3D [23])	Point Cloud	Image	High
Multistage (Frustum PointNet [25], Frustum ConvNet [26], Frustum-PointPillars [27])	Point Cloud Added Features	2D Detection Results	High
Ours (ExistenceMap-PointPillars)	Point Cloud	2D Detection Results	Low

**Table 2 sensors-23-08367-t002:** Number of objects per class.

	Test	Train	Total
Car	25,166	110,371	135,537
Bus	2006	4114	6120
Truck	5135	23,544	28,679
Trailer	1087	1776	2863
Pedestrian	8771	44,093	52,864
Cyclist	1944	6403	8347
Total	44,109	190,301	234,410

**Table 3 sensors-23-08367-t003:** Setting parameters.

Parameter	Value
Point cloud range x [m]	(−79.36, 79.36)
Point cloud range y [m]	(−79.36, 79.36)
Point cloud range z [m]	(−2.00, 4.00)
Pillar size [m] (Length × Width × Height)	0.32 × 0.32 × 6.00
Max number of pillars (train, test)	(16,000, 40,000)
Max number of points in a pillar	32
Pseudo image seize $\left(H,W,C\right)$	496 × 496 × 64
Pseudo 2D map size $\left(H,W,C\right)$	496 × 496 × 64
Epoch	160

**Table 4 sensors-23-08367-t004:** Anchor box size.

Class	Anchor Box Size [m] (Length× Width× Height)
Car	4.73 × 2.08 × 1.77
Bus	9.00 × 2.30 × 3.20
Truck	9.00 × 2.30 × 3.20
Trailer	9.30 × 2.30 × 3.20
Pedestrian	0.91 × 0.84 × 1.74
Cyclist	1.81 × 0.84× 1.77

**Table 5 sensors-23-08367-t005:** Evaluation results on our test dataset by AP (%). AP means Average Precision. PP, F-PP, and EM-PP stand for PointPillars [12], Frustum-PointPillars [27], and ExistenceMap-PointPillars.

Method	mAP	AP					
		Car	Bus	Truck	Trailer	Pedestrian	Cyclist
PP	39.06	67.80	58.69	43.48	23.37	13.16	27.88
F-PP	29.26	58.46	49.30	31.28	19.10	6.61	10.85
F-PP only Filter	29.58	58.92	49.42	31.02	21.86	5.95	10.33
F-PP only Weight	35.07	66.08	56.67	39.78	24.60	8.36	14.95
EM-PP trained only Camera	42.01	**68.73**	59.21	**48.65**	**29.82**	14.30	31.34
EM-PP trained only Tracking	41.87	68.72	57.73	45.60	23.61	14.98	40.57
EM-PP trained Camera & Tracking	**43.25**	68.60	**60.89**	46.16	25.27	**15.88**	**42.69**

**Table 6 sensors-23-08367-t006:** Evaluation results on our test dataset by AP (%). AP means Average Precision. PP and EM-PP stand for PointPillars [12] and ExistenceMap-PointPillars. Each ExistenceMap-PointPillars is trained with camera and tracking information.

Method	mAP	AP					
		Car	Bus	Truck	Trailer	Pedestrian	Cyclist
PP	39.06	67.80	58.69	43.48	23.37	13.16	27.88
EM-PP (no prior data in testing)	39.98	67.32	59.65	45.38	21.74	13.33	32.44
EM-PP	**43.25**	**68.60**	**60.89**	**46.16**	**25.27**	**15.88**	**42.69**

**Table 7 sensors-23-08367-t007:** Evaluation results on our test dataset by precision (%) and recall (%). The score threshold is set at 0.25. PP and EM-PP stand for PointPillars [12] and ExistenceMap-PointPillars. Each ExistenceMap-PointPillars is trained with camera and tracking information.

**Method**	**Car**		**Bus**		**Truck**	
	**Precision**	**Recall**	**Precision**	**Recall**	**Precision**	**Recall**
PP	58.91	78.14	66.51	57.73	46.97	**22.08**
EM-PP (no prior data in testing)	**63.71**	76.42	76.20	**57.78**	**55.00**	18.95
EM-PP	61.36	**78.62**	**77.38**	57.48	54.67	21.07
**Method**	**Trailer**		**Pedestrian**		**Cyclist**	
	**Precision**	**Recall**	**Precision**	**Recall**	**Precision**	**Recall**
PP	46.97	53.32	31.45	21.29	47.68	36.99
EM-PP (no prior data in testing)	**55.00**	52.93	38.63	18.86	58.04	34.16
EM-PP	54.67	**54.22**	**39.36**	**23.40**	**59.78**	**49.85**

**Table 8 sensors-23-08367-t008:** Evaluation results on our test dataset by precision (%) and recall (%) inside and outside of error ellipse. The score threshold is set to 0.25. PP and EM-PP stand for PointPillars [12] and ExistenceMap-PointPillars. Each ExistenceMap-PointPillars is trained with camera and tracking information.

Area	Method	Cars		Pedestrian		Cyclist	
		Precision	Recall	Precision	Recall	Precision	Recall
Inside of error ellipse	PP	62.38	76.86	45.10	25.75	**74.06**	48.48
	EM-PP	**64.69**	**77.49**	**46.62**	**31.46**	69.57	**69.51**
Outside of error ellipse	PP	30.99	**39.49**	20.88	**16.51**	19.78	19.08
	EM-PP	**37.99**	38.95	**29.04**	14.76	**33.33**	**19.21**

**Table 9 sensors-23-08367-t009:** Processing time on NVIDIA A100-SXM4-4-GB.

Method	Processing Time (ms)					
	Pillar Feature Net	Existence Map Feature Net	Feature Agg	Backbone	Detection Head	Total
PP	76.09	-	-	2.43	0.93	79.46
F-PP	18.79	-	-	2.46	0.88	22.13
EM-PP	76.74	1.31	0.06	2.40	1.06	81.58

**Table 10 sensors-23-08367-t010:** Evaluation results of ablation study for data augmentation on our test dataset by precision (%) and recall (%). The score threshold is set to 0.25. PP and EM-PP stand for PointPillars [12] and ExistenceMap-PointPillars. Each ExistenceMap-PointPillars is trained with camera and tracking information. In the test processing, ExistenceMap-PointPillars given prior data (upper line) and not given (bottom line) are prepared. Cars include cars, buses, trucks, and trailers.

	Method	Cars		Pedestrian		Cyclist		All Class
		Precision	Recall	Precision	Recall	Precision	Recall	F-Score
Given prior data in the testing	EM-PP w/o FN aug	**70.94**	45.20	**40.68**	10.30	**59.82**	35.08	48.52
	EM-PP w/o extreme FN aug	61.68	70.97	39.37	22.83	57.45	50.36	59.74
	EM-PP w/o FP aug	60.68	**71.96**	40.10	22.84	57.18	**51.23**	59.77
	EM-PP	61.19	71.72	39.36	**23.40**	59.78	49.85	**59.88**
No prior data in the testing	EM-PP w/o FN aug	62.82	2.40	0.00	0.00	0.00	0.00	3.53
	EM-PP w/o extreme FN aug	**63.96**	68.75	**39.51**	18.29	54.20	**35.85**	59.19
	EM-PP w/o FP aug	63.17	**69.83**	38.98	18.69	53.99	34.77	**59.28**
	EM-PP	63.10	69.82	38.63	**18.86**	**58.04**	34.16	**59.28**

## Data Availability

Data sharing not applicable.

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
