# Peer review of "ExistenceMap-PointPillars: A Multifusion Network for Robust 3D Object Detection with Object Existence Probability Map [Author-notes fn1-sensors-23-08367]"

_sensors, 2023, doi:10.3390/s23208367_

Round 1

Reviewer 1 Report

1.     Check the English presentation of some long sentences.

2.     Improve your introduction. First, you can divide it into two parts, including introduction and related work. Second, authors need to add more references and detailed descriptions of them.

3.     What are the limitations of the proposed model? Please add the relevant contents in the conclusion section.

4.     Table 1 lacks explanation, Table 2 is unnecessary.

5.     The paper's formatting is highly irregular and requires adjustments. For instance, Figure 1 is excessively distant from its corresponding description. Furthermore, Figures 4, 9, and Table 10 lack proper alignment, as their titles and content do not share the same page. Consider relocating Figures 6-7 to section 3.1.1. Notably, line 147 consists of just one word, and lines 95 to 100 lack the appropriate indentation. These issues warrant rectification for enhanced readability and cohesiveness.

Author Response

I would like to thank the reviewer for careful reading and valuable comments. We revised the paper based on the comments. Please see the attachment.

Reviewer 2 Report

The manuscript introduces the ExistenceMap-PointPillars method and how it improves object recognition for safe autonomous driving.

In my opinion, the manuscript needs to be carefully revised and improved in the following aspects:

1.It is recommended that the content of the manuscript should not exceed the layout.

2.The basis of sensor fusion is spatio-temporal matching. How do the methods described in the manuscript achieve consistency in time and space?

3.It is recommended to explain the interconversion theory of 2D and 3D data. Does this operation cause loss of accuracy?

4.The experiment used the author's own data set. How to determine the data set label?

5.It is recommended to compare and analyze the following articles separately in a targeted manner.

Object Classification using CNN-Based Fusion of Vision and LIDAR in Autonomous Vehicle Environment. DOI: 10.1109/TII.2018.2822828.

A Structure Constraint Matrix Factorization Framework for Human Behavior Segmentation. DOI: 10.1109/TCYB.2021.3095357.

An Interacting Multiple Model for Trajectory Prediction of Intelligent Vehicles in Typical Road Traffic Scenario. DOI: 10.1109/TNNLS.2021.3136866.

Robust Lateral Trajectory Following Control of Unmanned Vehicle Based on Model Predictive Control. DOI: 10.1109/TMECH.2021.3087605.

It is recommended to check the full text carefully.

Author Response

(The authors gave the same response as above.)

Round 2

Reviewer 1 Report

The authors have resolved all my doubts, so I suggest accepting this manuscript.